# Assessing the Livelihood Vulnerability of Herders to Changing Climate in Chui Oblast, Kyrgyz Republic

Aliya Ibraimova [1], Woo-Kyun Lee [2], Murat Zhumashev [1] and Sonam Wangyel Wang [3],*

[1] CAMP Alatoo Public Foundation, Bishkek 720031, Kyrgyzstan; aliya@camp.kg (A.I.); murat@camp.kg (M.Z.)
[2] Division of Environmental Science and Ecological Engineering, Korea University, Seoul 02841, Republic of Korea; leewk@korea.ac.kr
[3] OJEong Resilience Institute, Division of Environmental Science and Ecological Engineering, Korea University, Seoul 02841, Republic of Korea
* Correspondence: wangsonam@korea.ac.kr

**Abstract:** Kyrgyzstan is a country that is heavily reliant on agricultural products and animal husbandry, making it particularly vulnerable to the effects of climate change. Using the livelihood vulnerability index (LVI), this study attempted to assess vulnerability to climate change impacts on land, biodiversity, herders, and livestock in Kyrgyzstan's mountainous areas. The survey was conducted using respondent interviews. In the Chon-Kemin valley target area, we interviewed 198 nomad households with livestock. The study found that communities rely heavily on pasture resources, that there are almost no human–wildlife conflicts (HWCs), and that climate disasters have limited impact. Major components of herders' sensitivity and adaptive capacity resulted in low numbers. This, combined with rare climate disasters such as droughts and thus low exposure, reduced vulnerability under the IPCC framework. However, any increase in climate disasters, combined with continued land-use pressure, will severely impact herders and their livelihoods. In Kyrgyzstan, hundreds of livestock and dozens of wild animals have already died as a result of late frosts in pastures. National policies and programs should be revised to improve adaptive capacity. This study calls into question the importance of improving wildlife research and capacity building, as well as cooperation between local and national stakeholders.

**Keywords:** livelihood vulnerability index; Kyrgyzstan; Third Pole; climate change; herders





## 1. Introduction

Climate change, biodiversity loss, and livelihoods are interconnected in a complex and multifaceted way. Biodiversity loss refers to the rapid decline in the variety and abundance of plant and animal species worldwide, driven by habitat destruction, overexploitation, pollution, and climate change. This affects people's livelihoods, including activities such as farming, fishing, and forestry [1].

The impacts of climate change and biodiversity loss on livelihoods are closely intertwined [2]. Many of the world's poorest people depend on natural resources, such as pastures, forests, fisheries, and agriculture, for their livelihoods. Climate change and biodiversity loss threaten these resources, making it harder for people to earn a living and provide for their families. For example, changes in temperature and rainfall patterns can reduce crop yields, increase the number of pests and diseases, cause flooding, and damage agricultural lands. Loss of biodiversity can also disrupt the balance of ecosystems and reduce the availability of food and other resources that people depend on [3].

In addition to the direct impacts on livelihoods, climate change and biodiversity loss can have broader social and economic consequences. For example, natural disasters and environmental degradation can lead to displacement, migration, and conflicts over resources, all of which can have far-reaching impacts on communities and societies. These impacts are often felt most acutely by marginalized and vulnerable populations, including

indigenous peoples, women, and children [4]. Addressing climate change and biodiversity loss requires a multifaceted approach that takes into account the interconnections between these issues and the social and economic factors that drive them. This includes efforts to reduce greenhouse gas emissions, protect and restore ecosystems and biodiversity, and support sustainable livelihoods and community resilience [5,6].

One method to better understand these complex connections is through a livelihood vulnerability assessment. LVI studies can offer empirical guidance for policymakers and managers to develop programs and actions to reduce vulnerability and develop plans for climate adaptation [7]. According to the IPCC (2007), vulnerability is a function of adaptive capacity, sensitivity, and exposure [8], and many researchers have developed indicators by combining biophysical indicators with socio-economic characteristics into a composite index, such as the livelihood vulnerability index [9,10]. LVI studies yield specific insights to understand which areas are affected the most [10], thereby greatly aiding policymakers and managers to target adaptation and vulnerability reduction interventions [11].

Pastoralists are of the utmost significance in the context of Kyrgyzstan due to its mountainous landscape, which makes other forms of agriculture more difficult. Their livelihoods depend directly on the availability of natural resources. However, natural resources are under continuous degradation from anthropogenic influences exacerbated by climate change, making adaptation extremely difficult. One group of people who are most affected by climate change are herders, who live in harsh weather conditions in the remote mountains of Kyrgyzstan. The herders live in fragile and ecologically sensitive mountain landscapes that are vulnerable to the slightest change in the environment. However, no studies have been undertaken to assess their vulnerability to climate change. Our study aimed to understand the vulnerability of the herders living in the Chui Oblast region of Kyrgyz Republic (Figure 1). We used the sustainable livelihood approach (SLA) and the LVI developed by Hahn et al. (2009) [10] to determine the vulnerability of the herders in the study area.

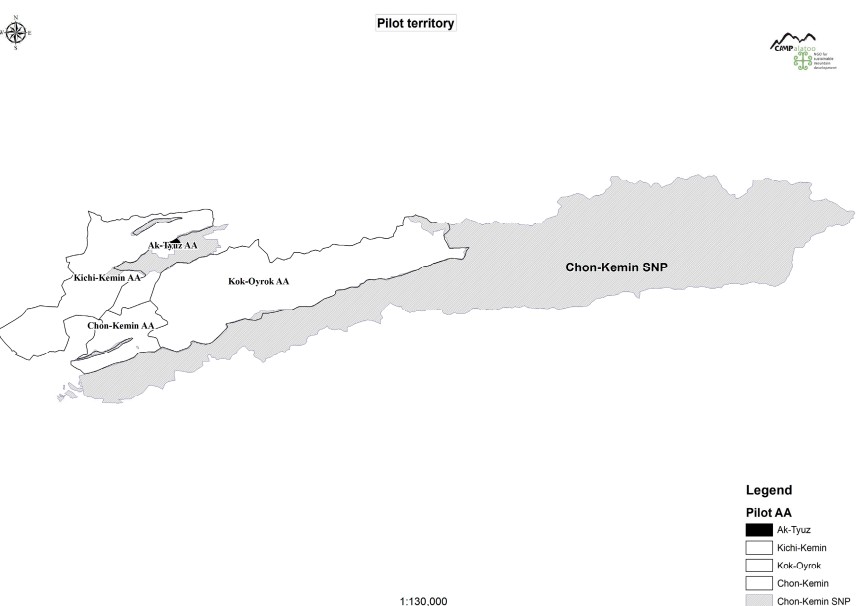

**Figure 1.** Map of the Chon Kemin SPN and showing study areas.

The SLA assesses the vulnerability of household assets to shocks and trends, as well as seasonality [12] Vulnerability is defined by the SLA as "a lack of resilience to changes that threaten welfare, which can take the form of sudden shocks, long-term trends, or seasonal cycles [that] typically bring increasing risk and uncertainty" [13]. The magnitude of the sustainable livelihood approach (SLA) evaluates the susceptibility of household assets to shocks, trends, and seasonality [12].

According to the IPCC Third Assessment Report (2007) [14,15], (IPCC 2007), vulnerability of the system is a positive result of its exposure and sensitivity, but a negative result of its adaptability. The type and extent to which a system is exposed to a significant shift in climate is referred to as exposure. The degree to which a system is affected, either negatively or positively, by climate-related stimuli is referred to as sensitivity. The ability of a system to adjust to climate fluctuation and extremes in order to avoid potential damage or cope with its consequences is referred to as adaptive capacity. The LVI and the LVI-IPCC developed by Hahn et al. were used in this study to calculate the risks from climate variability [10]. Many other researchers have successfully used this approach to assess vulnerability. The severity of vulnerability is determined by external threats to people's, households', and communities' well-being [12].

## 2. Materials and Methods

Kyrgyzstan is a mountainous country with a total land area of 200,000 km$^2$, 49% of which is covered by pastures [16]. Agriculture employs more than 60% of the population (farming and livestock rearing). The elevation of the country ranges from 500 m at its lowest point to 7134 m at the highest mountain's peak. Kyrgyzstan is bordered to the east and southeast by China, to the north by Kazakhstan, to the west by Uzbekistan, and to the south by Tajikistan [17].

The pilot Chui region is located in Kyrgyzstan's northwestern region. It shares borders with Kazakhstan to the north and west. The Chui and Chon-Kemin valleys, as well as the slopes of the Kyrgyz Zaili and Kyungey Ala-Too mountains, are all part of the region. The region contains a wide range of ecosystems, from semi-deserts to the nival belt, and it has diverse biological and landscape diversity. The national park located in the region is situated between 1500 and 4771 m above the sea level [18]. The Chui region consists of eight districts, including Kemin district, which contains the pilot areas. The district has a population of over 44,118. There are 12 Ayil Akmak, also known as short AA, which are administrative territorial units made up of 1 or more village districts. Four of them, Kok-Oyrok, Chon-Kemin, Duisheyev, and Aktyuz AAs, border the natural park where the study was conducted (Figure 1). The park was established in 1997 with the goal of preserving natural biological and landscape diversity for tourism and environmental education development [18].

As shown in Table 1, the AA composition ranged from one to five villages, with populations ranging from 697 to 4185 people. Kyrgyz make up nearly the entire population. The main types of economic activities, such as livestock rearing and agriculture, are defined by natural conditions. Because these settlements are close to the capital, no significant migration was observed, apart from Ak-Tyuz, which has a population made up mostly of the elderly and children. There are no pastures or agricultural lands assigned to Ak-Tyuz households because it was once an industrial settlement with only factory workers living there. However, after the factory closed in the 1990s, the villagers turned to livestock breeding. They only graze their livestock in the near-settlement pastures owned by the park and other AAs. As a result, the majority of Ak-Tyuz's youth migrated [19].

The region's livestock rearing situation (cattle, sheep, and horses) is generally developed. This can also be seen from the vast pasture areas available to the households of the three pilot AAs (Table 1). According to the available information available in the literature and according to our interviews, human–wildlife conflict does not exist in the area. There may be wolf attacks, but under current legislation, it is legal to hunt them. Snow leopards are said to have a habitat in the pilot area, but they have not been seen in decades, according to stakeholder interviews. Potatoes, cereals, sugar beets, and perennial forage grasses are the most important crops grown. Horticulture (limited to household plots) and beekeeping are poorly developed [18].

**Table 1.** Characteristics of the study area.

| Ayil Aimak | Villages | Area | | Population Size (People) |
|---|---|---|---|---|
| Ak-Tyuz | Ak-Tyuz | No agricultural lands | | 697 |
| Kok-Oirok | Kaiyndy Tegirmenti Karool-Dobo | - Irrigated arable land: 1719 ha; <br> - Rainfed arable land: 553 ha; <br> - Hayfields: 373 ha; <br> - Pastures: 24,901 ha. | | 4185 |
| Duisheevsky | Kichi-Kemin | - Irrigated arable land: 2001 ha; <br> - Pastures: 9952 ha. | | 3056 |
| Chon-Kemin | Kyzyl-Bayrak Shabdan Kalmak-Ashuu Tar-Suu Tort-Kul | - Irrigated arable land: 1621 ha; <br> - Rainfed arable land: 81 ha; <br> - Hayfields: 802 ha; <br> - Pastures: 18,515 ha. | | 4097 |

The deeply incised Chon Kemin valley is located in the Kyrgyz part of the Tien Shan mountains, on the border with Kazakhstan and between the Zailiyskiy and Kungey Alatau ranges. The park is divided into several local areas, the largest of which is located in the Chon-Kemin River basin and covers 114,362 hectares (92.5 percent).

The Chui fauna is part of the Western Tenir-Too zoogeographical area. There are over 360 vertebrate species, including over 15 fish species, 280 bird species, and 50 mammal species [18].

The biodiversity of the specially protected natural area (SPNA) includes eleven Red List bird species and six mammals, including brown bear (*Ursus arctos*), Eurasian lynx (*Lynz lynx*), snow leopard (*Panthera uncia*), argali (*Ovis ammon*), and others. Flora is under-studied, but it contains about 650 species. The Chon-Kemin basin contains 109 glaciers. Zhangyryk (8.9 km), Southern Zhangyryk (8.0 km), At-Jailoo (7.6 km), and Novy (6.4 km) are the largest glaciers. Because of rising temperatures, glaciers are receding. The climate has warmed over the last 20 years (Figure 2), resulting in an increase in the intensity of extreme weather events, such as heat waves, droughts, and so on [20].

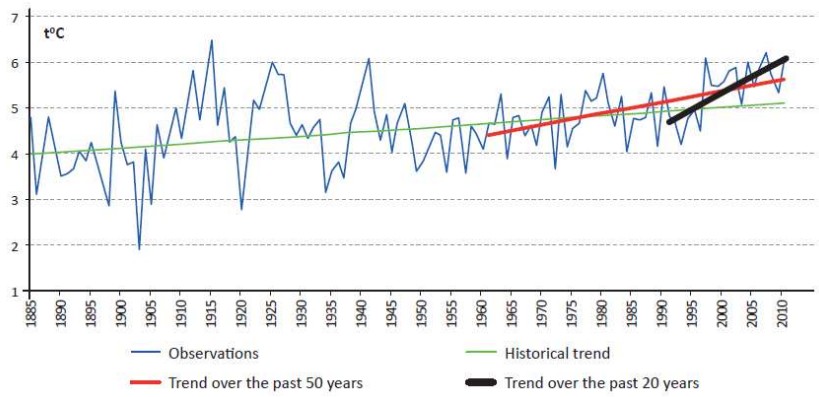

**Figure 2.** The change trend in average annual temperature in Kyrgyzstan. Source: Climate profile of the Kyrgyz Republic, cited in the Third National Communication (2016) [20].

The northern slopes are colder and wetter than the southern slopes, which are warmer and drier. The general climatic factors associated with the area's geographic latitude, which affect the climate of the region as a whole, are supplemented here by many local microclimate factors. As the valley gradually rises in the direction of the Kyrgyz ridge, the amount of precipitation increases. According to climatic data obtained for the period 2007–2021, the average precipitation for the entire Chui region was around 1000 mm per year. Between 2007 and 2021, the maximum temperature in July and August was 25–26 degrees Celsius, and the minimum temperature was −9–10 degrees Celsius in January and February [21].

### 2.1. Data Collection

To answer research questions and achieve study objectives, the study used a multifaceted approach. The following methods were used to collect both qualitative and quantitative data: (i) a desktop review; (ii) household interviews and field observations; and (iii) consultative meetings with experts and policymakers.

The research team validated information, gathered additional materials, and met with SPNA, Aiyl Okmotu (short for Kyrgyz local self-administration), and pasture committee representatives. The collected data were used to validate, supplement, and correct the survey findings (Figure 3).

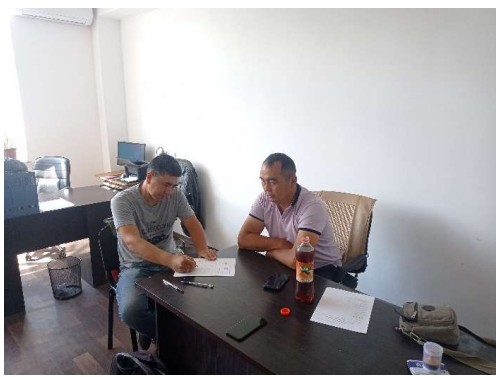 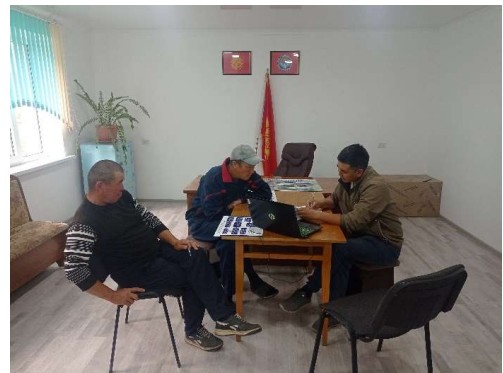

**Figure 3.** Interviews with Aiyl Okmotu heads and pasture committee.

As there was no way to obtain specific climate data on the areas, historical data indicating changes in temperature, precipitation, and extreme weather events were gathered from FAO open sources.

In June 2022, 198 households were surveyed to collect primary data. We used a pretested, semi-structured questionnaire to gather the information needed to create an LVI that can be embedded in the SLA. The survey questionnaires were pre-tested in the field in Bhutan before being slightly adapted and clarified for Kyrgyzstan's conditions. The pre-test enabled the Kyrgyz team to estimate interview time, identify potential barriers, and revise the questionnaire. The final questionnaire gathered data on exposure, sensitivity, adaptive capacity, perceptions, and adaptation strategies. Through desktop review, pretesting, expert consultation, and the study purposes, 53 sub-components or indicators were defined based on these 14 components for the 5 assets. Human, social, natural, physical, and financial assets are the five types of assets. Socio-demographic profile; social networks; health; agriculture and food security; livelihood strategies; human–wildlife conflicts; finance and income; infrastructure; housing type; natural resources; energy; land, water, and sanitation; natural disasters; and climate variability are the 14 major components. Respondents were chosen from a list of livestock-owning households. These are the lists provided by the AOs. The head of the household was interviewed in Kyrgyz and Russian using the pre-tested questionnaire. When the head of the household could not be reached, the spouse was interviewed. Each interview lasted about 15–20 min. To make the team's work easier, the questionnaire was integrated into the Kobo Toolbox data collection tool. The gathered data were encoded into an Excel database, double checked, and cleaned up for analysis in Excel.

A total of ten villages, representing four AAs, were interviewed. Respondents from one to five villages are included in each pilot AA.

Because livestock grazing implies a nomadic way of life, almost all herders were men (Table 2). The proportion of women polled represented female members of herding families. The number of interviewed herders was very small in Ak-Tyuz Aiyl Aimak, which has a population of only 697 people and, as previously stated, no pastures.

**Table 2.** Demographic information of survey respondents.

| AA | Gender | | Age | |
|---|---|---|---|---|
| | **M** **% n** | **F** **% n** | **$\geq 18 \leq 65$** **% n** | **>65** **% n** |
| Ak-Tyuz | 83 (5) | 17 (1) | 50 (3) | 50 (3) |
| Duisheev | 100 (12) | 0 (0) | 92 (11) | 8 (1) |
| Kok-Oirok | 90 (72) | 10 (8) | 94 (75) | 6 (5) |
| Chon-Kemin | 86 (86) | 14 (14) | 96 (96) | 4 (4) |
| Total | 88.38 (175) | 11.61 (23) | 93.43 (185) | 6.57 (13) |

*2.2. Vulnerability Analysis*

The data were analyzed in two major steps: (i) determination of the weighted-average vulnerability, or LVI; and (ii) determination of the LVI using the IPCC framework. To calculate the livelihood vulnerability indices, data were entered and analyzed using the Kobo toolbox and MS Excel.

The Hahn et al. [10] method was used to calculate the livelihood vulnerability index. The LVI had 14 major components (Table 3). Each of the 53 sub-components was calculated as equally contributing to the overall index of the 14 major components using the LVI balanced weighted-average approach [10] (Sullivan et al., 2002 [22], Table 2). Each major component may have a varying number of sub-components. The LVI calculation process is divided into four major stages:

(1) Convert raw field data into quantifiable units, such as ratios, percentages, and indices;
(2) Using Equation (1), standardize each sub-component as an index so that when measured on a different scale, all measures can be combined into an overall LVI.

$$Index_{S_k} = \frac{S_k - S_{min}}{S_{max} - S_{min}} \tag{1}$$

(3) Sub-components were averaged using Equation (2) to calculate the value of each major component at this stage [10]. For example, the average scores for sub-components, such as family members, working members, level of education, age, and gender, resulted in the score for the major component socio-demographic profile.

$$M_k = \frac{\sum_{i=1}^{n} S_{ki}}{n} \tag{2}$$

where $M_k$ = 1 of the 14 major components of the study (Table 2). Index $S_{ki}$ represents the sub-components indexed by *i*, which comprise each major component; and n is the number of sub-components in each major component. The major components are ranked from 0 (least vulnerable) to 1 (most vulnerable). Finally, in stage 4, the LVI for each group is computed by combining the weighted average of all major components using Equation (3). The LVI ranges from 0 (least vulnerable) to 1 (most vulnerable).

$$LVI_k = \frac{\sum_{i=1}^{6} W_{mi} M_{ki}}{\sum_{i=1}^{9} W_{mi}} \tag{3}$$

where $LVI_k$ is the study group's livelihood vulnerability index equaling the weighted average of the 14 major components. The weight of each major component, $W_{mi}$, is the number of sub-components that comprise each major component and are all included to ensure that each sub-component contributes equally to the overall LVI [22] (Sullivan 2002).

**Table 3.** Summary of the LVI results for 14 components in Ak-Tyuz, Kok-Oirok, Duisheevsky, and Chon-Kemin AAs; major components; major component values; number of sub-components.

| Major Components | Major Components Values | | | | Number of Sub-Components | |
|---|---|---|---|---|---|---|
| | **Ak-Tyuz** | **Kok-Oirok** | **Duisheevsky** | **Chon-Kemin** | | |
| Socio-demographic | 0.39 | 0.04 | 0.03 | 0.02 | 3 | Adaptive capacity |
| Social networks | 0.60 | 0.44 | 0.63 | 0.43 | 5 | Adaptive capacity |
| Livelihood | 0.15 | 0.14 | 0.21 | 0.13 | 8 | Adaptive capacity |
| Finance and income | 0.33 | 0.18 | 0.33 | 0.17 | 2 | Adaptive capacity |
| Infrastructure | 0.22 | 0.25 | 0.30 | 0.30 | 2 | Adaptive capacity |
| Housing type | 0.33 | 0.53 | 0.00 | 0.09 | 1 | Adaptive capacity |
| Health | 0.27 | 0.19 | 0.10 | 0.13 | 2 | Sensitivity |
| Agriculture and food Security | 0.19 | 0.17 | 0.16 | 0.19 | 4 | Sensitivity |
| Human–wildlife conflict | 0.00 | 0.12 | 0.00 | 0.18 | 2 | Sensitivity |
| Natural resources | 1.00 | 0.98 | 0.92 | 0.98 | 1 | Sensitivity |
| Energy | 0.22 | 0.32 | 0.25 | 0.29 | 3 | Sensitivity |
| Land | 0.58 | 0.21 | 0.52 | 0.14 | 4 | Sensitivity |
| Water and sanitation | 0.33 | 0.24 | 0.29 | 0.26 | 4 | Sensitivity |
| Climate change and disasters | 0.06 | 0.13 | 0.01 | 0.17 | 8 | Exposure |
| Overall livelihood vulnerability index | 0.28 | 0.22 | 0.25 | 0.21 | 49 | |

### 2.3. Calculating LVI Based on IPCC Framework

The IPCC-LVI was calculated using the IPCC framework, which defines vulnerability as a positive function of exposure and sensitivity, as well as a negative function of adaptive capacity [10,14,15] (IPCC, 2007). In this approach, the study classified all the 14 major components into 3 categories of exposure, sensitivity, and adaptive capacity. The study population's exposure (E) was calculated based on the frequency of disasters (floods, landslides, etc.) and climate variability, which was calculated using the maximum and minimum monthly temperatures and precipitation (Equation (4)). Adaptive capacity (AC) was assessed using demographic profiles (such as female-headed households), different types of livelihood strategies (occupations such as civil servants, laborers, business owners, etc.), incomes, and the strength of social networks (Equation (5)). Finally, the current state of water, sanitation, and health was used to calculate sensitivity (S) (Equation (6)).

$$E_k = \frac{W_{e1}ND + W_{e2}CV}{W_{e1} + W_{e2}} \quad (4)$$

where $W_{c1}$ is the weight of natural disasters and $W_{c2}$ is the weight of climate variability, both of which are equal to the total number of sub-components. We also hypothesized that higher rates of climate variables and natural disasters would expose households to more extreme weather and climate change.

$$AC_k = \frac{W_{a1}SDP + W_{a2}LS + W_{a3}SN + W_{a4}K + W_{a5}I}{W_{a1} + W_{a2} + W_{a3} + W_{a4}W_{a5}} \quad (5)$$

where $W_{a1}$, $W_{a2}$, $W_{a3}$, $W_{a4}$, and $W_{a5}$ are weights of social demographic profile, livelihood strategy, social network, knowledge, and income, respectively.

$$S_k = \frac{W_{s1}H + W_{s2}F + W_{s3}S + W_{s4}W}{W_{s1} + W_{s2} + W_{s3} + W_{s4}} \quad (6)$$

where $W_{s1}$, $W_{s2}$, $W_{s3}$, and $W_{s4}$ are the weights of health, food, sanitation, and water, respectively.

The final IPCC-LVI was calculated by combining the weighted averages of exposure, sensitivity, and adaptive capacity in Equation (7).

$$\text{IPCC} - \text{LVI} = (E_k + AC_k) * S_k \tag{7}$$

The IPCC-LVI was scaled from −1 (least vulnerable) to 1 (most vulnerable).

## 3. Results

### 3.1. Overview

The results of the LVI values of 14 major components are represented collectively in Table 3, the spider diagram (Figure 4), and the vulnerability triangle diagram (Figure 5) using a scale of 0.10 units, with 0 being the least vulnerable in the center and 1.00 being the most vulnerable, shown on the figure's outer frame. The results for Ak-Tyuz, Kok-Oirok, Duisheevsky, and Chon-Kemin were derived from 49 sub-components. The total LVI demonstrates that the studied herders and related AAs are not particularly vulnerable. Overall, the LVI differs only slightly between Ak-Tyuz (0.28), Kok-Oirok (0.22), Duisheevsky (0.25), and Chon-Kemin (0.21), with Ak-Tyuz and Duisheevsky having a slightly higher LVI than Kok-Oirok and Chon-Kemin. The diagram also shows that Ak-Tyuz and Duiusheevsky have the highest vulnerability in most components, with the exception of the component housing type, where Kok-Oirok has the highest vulnerability (0.53). All of the studied village districts have high vulnerability to natural resources (between 0.92 and 1.00), as well as moderate vulnerability to social networks (between 0.43 and 0.63). Land-use vulnerability is also higher in the Ak-Tyuz and Duisheevsky village district (Ak-Tyuz 0.58; Duisheevsly 0.52). Table 4 shows the detailed LVI values for each major component and its sub-components. The LVI-IPCC values calculated for the four village districts are −0.09 for Ak-Tyuz, −0.02 for Kok-Oirok, −0.08 for Duisheevsky, and −0.01 for Chon-Kemin, as shown in Figure 5.

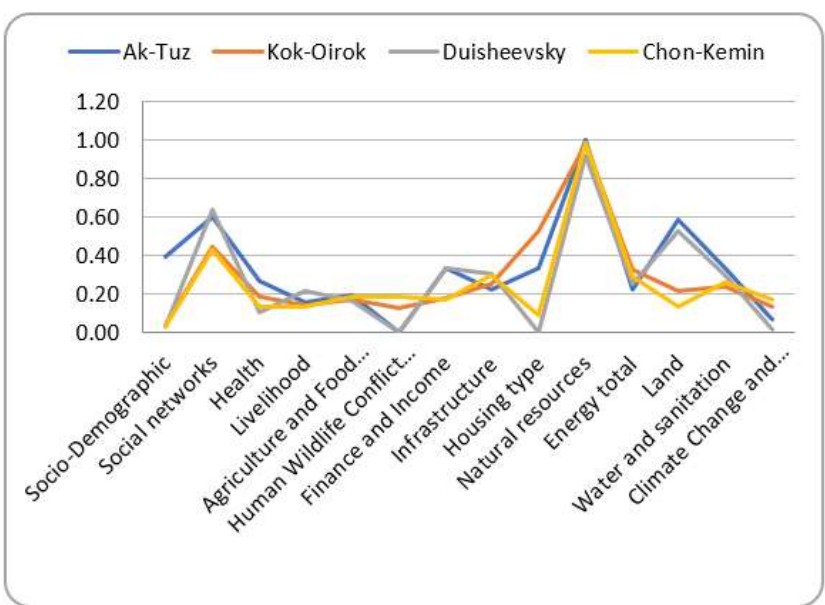

**Figure 4.** Calculated LVI components for Ak-Tyuz, Kok-Oirok, Duisheevsky, and Chon-Kemin Aiyl district.

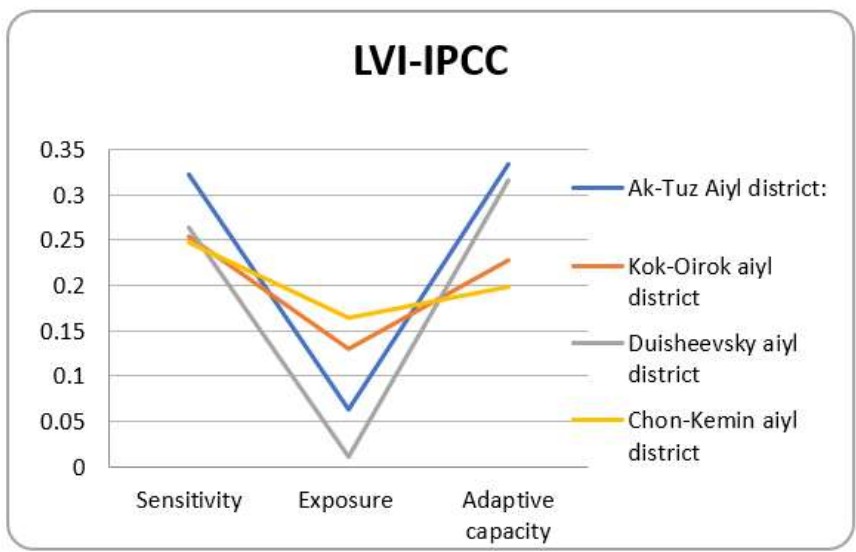

**Figure 5.** Vulnerability triangle diagram of the contributing factors of the livelihood vulnerability index—IPCC (LVI–IPCC) for Ak-Tyuz, Kok-Oirok, Duisheevsky, and Chon-Kemin AAs.

**Table 4.** Major components and sub-components calculation results of four AAs.

| Major Components | Sub-Components | Regions | | | |
|---|---|---|---|---|---|
| | | Ak-Tyuz | Kok-Oyrok | Duish Eevsky | Chon-Kemin |
| Socio-demographic | Population density | 1.00 | 0.03 | 0.03 | 0.02 |
| | Dependency ratio | 1.00 | 0.10 | 0.09 | 0.05 |
| | % of female-headed households | 0.17 | 0.01 | 0.00 | 0.02 |
| | % of household heads who did not attend school | 0.00 | 0.00 | 0.00 | 0.00 |
| Social networks | % of households who have received information on climate change and disasters | 1.00 | 0.51 | 0.75 | 0.36 |
| | % of households who received support from the government | 0.00 | 0.03 | 0.08 | 0.07 |
| | % of households who are members of any community-based groups | 0.00 | 0.19 | 0.33 | 0.20 |
| | % of households who have a mobile phone | 1.00 | 0.93 | 1.00 | 0.96 |
| | % of households who have radio at home | 1.00 | 0.56 | 1.00 | 0.55 |
| Health | % households that reported at least one person with chronic sickness | 0.50 | 0.25 | 0.17 | 0.05 |
| | Average distance to health centers | 0.03 | 0.12 | 0.03 | 0.22 |
| Agriculture and food security | % of households dependent solely on livestock rearing for food | 0.00 | 0.01 | 0.00 | 0.01 |
| | Average number of livestock owned | 0.00 | 0.04 | 0.01 | 0.04 |
| | % of households dependent on agriculture | 0.33 | 0.63 | 0.42 | 0.63 |
| | Average food insufficient months | 0.29 | 0.15 | 0.21 | 0.22 |
| | % of households that are food sufficient | 0.33 | 0.04 | 0.17 | 0.03 |

**Table 4.** *Cont.*

| Major Components | Sub-Components | Regions | | | |
|---|---|---|---|---|---|
| | | Ak-Tyuz | Kok-Oyrok | Duish Eevsky | Chon-Kemin |
| Livelihood strategy | % of households with member working outside the community | 0.00 | 0.00 | 0.00 | 0.00 |
| | Average agricultural livelihood diversity index | 0.40 | 0.38 | 0.56 | 0.38 |
| | % of household main income dependent on more than one family member employed | 0.50 | 0.51 | 0.83 | 0.49 |
| | % of household with members employed in civil service | 0.00 | 0.03 | 0.08 | 0.01 |
| | % of households with members engaged in business | 0.00 | 0.05 | 0.00 | 0.03 |
| | % of households with members engaged in transport (taxi drivers, truckers, bus, etc.) | 0.00 | 0.00 | 0.00 | 0.00 |
| | % of households with members engaged in daily wage work (e.g., working on road, construction, etc.) | 0.17 | 0.00 | 0.00 | 0.00 |
| HW Conflicts | % of households reporting loss of livestock to snow leopard | 0.00 | 0.00 | 0.00 | 0.00 |
| | % of households reporting loss of livestock to other predators | 0.00 | 0.00 | 0.00 | 0.00 |
| | % of households reporting loss of livestock to climate and weather | 0.00 | 0.20 | 0.00 | 0.29 |
| Finance and Income | % of households who have debt to pay back to neighbors or financial institution | 0.50 | 0.18 | 0.50 | 0.23 |
| | % of households who have no access to loans | 0.17 | 0.18 | 0.17 | 0.11 |
| Infrastructure | Average time to reach the nearest vehicle station | 0.24 | 0.25 | 0.38 | 0.32 |
| | Average time to reach the nearest market | 0.20 | 0.25 | 0.22 | 0.27 |
| Housing Type | Average time to reach the nearest vehicle station | 0.24 | 0.25 | 0.38 | 0.32 |
| | Average time to reach the nearest market | 0.20 | 0.25 | 0.22 | 0.27 |
| Natural Resources | % of household that depend on natural resources | 1.00 | 0.98 | 1.00 | 0.98 |
| Energy | % of households using only forest-based energy for cooking | 0.00 | 0.04 | 0.00 | 0.07 |
| | % of household using livestock dung for cooking | 0.67 | 0.90 | 0.42 | 0.75 |
| | % of households reporting energy insufficiency | 0.00 | 0.03 | 0.33 | 0.04 |
| Land | % of households who owns pasture | 0.50 | 0.20 | 0.75 | 0.08 |
| | % of households reporting shortage of grazing land | 1.00 | 0.13 | 0.42 | 0.05 |
| | % of households reporting pastureland degradation by weather and climate in the last 15 years | 0.50 | 0.18 | 0.58 | 0.03 |
| | Reasons for pasture degradation | 0.33 | 0.34 | 0.33 | 0.38 |
| Water and Sanitation | % of households with access to clean drinking water | 1.00 | 0.91 | 1.00 | 0.98 |
| | % of households reporting water conflicts | 0.33 | 0.04 | 0.17 | 0.05 |
| | % of households with access to private toilets | 1.00 | 1.00 | 1.00 | 1.00 |
| | % of households reporting drinking water for livestock is not enough | 0.00 | 0.00 | 0.00 | 0.00 |
| Climate Change and Disaster | Average number of climate disasters reported in the past 5 years | 0.17 | 0.29 | 0.08 | 0.36 |
| | Average number of floods in the past 5 years | 0.00 | 0.08 | 0.00 | 0.02 |
| | Average number of wind/hailstorms in the last 5 years | 0.00 | 0.00 | 0.00 | 0.00 |
| | Average number of landslides in the past 5 years | 0.00 | 0.00 | 0.00 | 0.01 |
| | Average number of droughts in the past 5 years | 0.17 | 0.10 | 0.00 | 0.24 |
| | Average number of forest fires in the past 5 years | 0.00 | 0.00 | 0.00 | 0.00 |
| | % of households not receiving climate advanced warning | 0.17 | 0.48 | 0.00 | 0.50 |
| | % of households suffering death and being a victim due to a natural disaster in the last 5 years | 0.00 | 0.10 | 0.00 | 0.19 |
| | Standard deviation of average monthly temperature based on daily max and min temperature by month between xx–xx was averaged for each area | 1.47 (NA) | 1.47 (NA) | 1.47 (NA) | 1.47(NA) |
| | Standard deviation from monthly rainfall based on average max and min by month between xx–xx was averaged for each area | 26.47 (NA) | 26.47 (NA) | 26.47 (NA) | 26.47 (NA) |

*3.2. Sub-Components*

The major components, as well as their respective sub-components for Ak-Tyuz, Kok-Oirok, Duisheevsky, and Chon-Kemin, are presented in Table 4. Figure 4 demonstrates the main components visually.

Ak-Tyuz had the highest vulnerability in the sub-component's population density (1.00) and dependency ratio (1.00), percent of households who have received information on climate change and disasters (1.00), percent of households who have a mobile phone (1.00), percent of households who have a radio at home (1.00), and percent of households reporting grazing land shortages (1.00), percent of households with access to safe drinking water (1.00), percent of households reliant on natural resources (1.00), and percent of households with access to private toilets (1.00). It also showed higher vulnerability in the percent of households that use livestock dung for cooking (0.67) and the percent of households that owe money to neighbors or financial institutions (0.50).

Kok-Oirok was most vulnerable in terms of percent of households with access to private toilets (1.00), percent of households relying on natural resources (0.98), percent of households owning a mobile phone (0.93), percent of households having access to clean drinking water (0.91), and percent of households cooking with livestock dung (0.90). Other vulnerable sub-components included percent of households dependent on agriculture (0.63), percent of households with a radio at home (0.56), percent of households living in permanent housing (0.52), percent of households whose main income is dependent on more than one family member employed (0.51), and percent of households not receiving climate advance warnings (0.48). It was more vulnerable than other village districts in terms of the percentage of households reporting livestock loss due to climate and weather (0.20).

Duisheevsky was the most vulnerable in terms of the percentage of households that rely on natural resources (1.00), the percentage of households that have access to private toilets (1.00), the percentage of households that have a mobile phone (1.00), the percentage of households that have a radio at home (1.00), the percentage of households that have access to clean drinking water (1.00), the percentage of households that have more than one family member employed (0.83), and the percentage of households that own pasture (0.75). Other more vulnerable sub-components included the percent of households reporting pastureland degradation due to weather and climate in the last 15 years (0.58), the average agricultural livelihood diversity index (0.56), and the percent of households who owe money to neighbors or financial institutions (0.50).

Chon-Kemin had the highest vulnerability in terms of the percentage of households with access to private toilets (1.00), the percentage of households with access to clean drinking water (0.98), the percentage of households that rely on natural resources (0.98), the percentage of households with a mobile phone (0.96), and the percentage of households that use livestock dung for cooking (0.75). It also demonstrated greater vulnerability in terms of the percentage of households dependent on agriculture (0.63), the percentage of households with a radio at home (0.55), and the percentage of households not receiving climate advance warning (0.50). In comparison to the other three village districts, it demonstrated greater vulnerability in the average number of climate disasters reported in the previous 5 years (0.36), the percent of households reporting livestock loss due to climate and weather (0.29), and the average number of droughts in the previous 5 years (0.24).

Two subcomponents are vulnerable. The standard deviation of average monthly temperature based on daily maximum and minimum temperature by month between 1991–2021 was averaged for each area (1.47), and the standard deviation of monthly rainfall based on daily maximum and minimum temperature by month between 1991–2021 was averaged for each area (26.47). Both values were statistically insignificant and were therefore excluded from the overall LVI calculation.

The following Table 5 summarizes the main components. Table 6 demonstrates the IPCC-LVI according to each village district. Tables 7–9 show the respective values for each village district according to sub components of adaptive capacity (Table 7), sensitivity (Table 8) and exposure (Table 9), which will be further discussed in the following chapter.

**Table 5.** Overview of the major components calculation results for four AAs.

| Major Components | Regions | | | |
|---|---|---|---|---|
| | Ak-Tyuz | Kok-Oyrok | Duisheevsky | Chon-Kemin |
| Socio-demographic | 0.39 | 0.04 | 0.03 | 0.02 |
| Social networks | 0.60 | 0.44 | 0.63 | 0.43 |
| Health | 0.27 | 0.19 | 0.10 | 0.13 |
| Agriculture and food security | 0.19 | 0.17 | 0.16 | 0.22 |
| Livelihood strategy | 0.15 | 0.14 | 0.21 | 0.13 |
| HW conflicts | 0.00 | 0.12 | 0.00 | 0.18 |
| Finance and Income | 0.33 | 0.18 | 0.33 | 0.17 |
| Infrastructure | 0.22 | 0.25 | 0.30 | 0.30 |
| Housing type | 0.33 | 0.52 | 0.00 | 0.09 |
| Natural resources | 1.00 | 0.98 | 0.92 | 0.98 |
| Energy | 0.22 | 0.32 | 0.25 | 0.29 |
| Land | 0.58 | 0.21 | 0.52 | 0.14 |
| Water and sanitation | 0.33 | 0.24 | 0.29 | 0.26 |
| Climate change and disaster | 0.06 | 0.13 | 0.01 | 0.17 |
| **Livelihood Vulnerability Index (LVI)** | **0.28** | **0.22** | **0.25** | **0.21** |

**Table 6.** IPCC-LVI calculation for each village district.

| IPCC-LVI | (Exposure-Adaptive Capacity) $\times$ Sensitivity |
|---|---|
| Ak-Tyuz | −0.09 |
| Kok-Oirok | −0.02 |
| Duisheevsky | −0.08 |
| Chon-Kemin | −0.01 |

**Table 7.** Adaptive capacity of the four AAs and their respective sub-components.

| Adaptive Capacity | Socio-Demographic | Social Networks | Livelihood | Finance and Debts | Infra-structure | Housing Type | Own Pasture-land | Total |
|---|---|---|---|---|---|---|---|---|
| Nr. of Sub-Components | 3 | 8 | 5 | 2 | 2 | 1 | 1 | 22 |
| Ak-Tuz | 0.39 | 0.15 | 0.60 | 0.33 | 0.22 | 0.33 | 0.50 | 0.33 |
| Kok-Oirok | 0.04 | 0.14 | 0.44 | 0.18 | 0.25 | 0.53 | 0.20 | 0.23 |
| Duisheevsky | 0.03 | 0.21 | 0.63 | 0.33 | 0.30 | 0.00 | 0.75 | 0.32 |
| Chon-Kemin | 0.02 | 0.13 | 0.43 | 0.17 | 0.30 | 0.09 | 0.08 | 0.20 |

**Table 8.** Sensitivity of the four village districts with their respective sub-categories.

| Sensitivity | Health | Water and Sanitation | Agriculture and Food Security | Human–Wildlife Conflict | Natural Resources | Energy | Land | Total |
|---|---|---|---|---|---|---|---|---|
| Nr. of Sub-Components | 2 | 4 | 4 | 2 | 1 | 3 | 3 | 19 |
| Ak-Tyuz | 0.27 | 0.33 | 0.19 | 0.00 | 1.00 | 0.22 | 0.61 | 0.32 |
| Kok-Oirok | 0.19 | 0.24 | 0.17 | 0.12 | 0.98 | 0.32 | 0.21 | 0.25 |
| Duisheevsky | 0.10 | 0.29 | 0.16 | 0.00 | 0.92 | 0.25 | 0.44 | 0.26 |
| Chon-Kemin | 0.13 | 0.26 | 0.19 | 0.18 | 0.98 | 0.29 | 0.15 | 0.25 |

**Table 9.** Exposure of the four village districts with their respective sub-components.

| Exposure | Climate Change and Disasters | Total |
|---|---|---|
| Nr. of Sub-Components | 8 | 8 |
| Ak-Tuz | 0.06 | 0.06 |
| Kok-Oirok | 0.13 | 0.13 |
| Duisheevsky | 0.01 | 0.01 |
| Chon-Kemin | 0.17 | 0.17 |

## 4. Discussion

Figure 5 depicts the IPCC-LVI values for Ak-Tyuz, Kok-Oirok, Duisheevsky, and Chon-Kemin. Overall, the three components of vulnerability are very low, indicating low

sensitivity, limited adaptive capacity, and limited exposure. The final IPCC LVI values are negative as a result of these values. This can be interpreted as low overall vulnerability of the communities in the study area to the effects of climate change. Growing values in exposure or sensitivity indices, on the other hand, have an immediate negative impact on the overall LVI. Households are heavily reliant on natural resources and lack diverse sources of income, among other things. Livelihoods may suffer significantly as a result of more severe and explicit climate change impacts and increased exposure.

### 4.1. Adaptive Capacity

The adaptive capacity values for the four village districts range from 0.20 (Chon-Kemin) to 0.33 (Ak-Tyuz) (Table 4). The Ak-Tyuz village district has the highest adaptive capacity vulnerability ratings, particularly in socio-demographic aspects, because it has the largest population density and dependence rate among the four village districts. Ak-Tyuz also has the highest scarcity of grazing land and the fewest herders (only six people during the period of fieldwork). The reliance on social networks is relatively low, which might be attributed to the complete level of education acquired by all participants in this study. Households are nearly entirely headed by males, and the areas are close to the capital, resulting in little to no migration between rural districts and the city. Households' adaptation capacity in terms of knowledge and accessibility to communication methods may be considerably greater, but there is little to no help from the government or a prospective support group. This is especially true in the case of climate-change-related weather shifts or environmental disasters. When the local populace has faster access to information via social networks and information and communication technology, they can respond more quickly and become less vulnerable [23–25]. Because of a lack of variety in livelihood options, the adaptation capacity of livelihoods in the four village districts is quite low. Diversification of livelihoods is essential for household survival [26]. According to research from South Asia, farmers diversify their livelihood earnings more when natural disasters strike more frequently [27]. When it comes to cash and debts, both Ak-Tyuz and Duisheevksy are fairly weak (both show 0.50 percent of households who have debts). Most families borrow money from individuals rather than financial organizations due to a lack of access to suitable loans. The state of infrastructure varies greatly: Some local herders can drive to the next village in 30 min, while others need 6 h. When driving is not an option, the next settlement may be reached on horseback in 2 to 18 h, and the same is true for the nearest market. These extended wait periods for commodities and public services increase sensitivity to climate-related concerns [28]. Because all herders have permanent dwellings in communities, the vulnerability of housing types solely applies to the housing situation in the pastures. Kok-Oirok had the most vulnerable housing situation (0.52), whereas Duisheevsky had none at all (0.00). One likely explanation is the position and accessibility of the pastures; easier access affords more dwelling alternatives. The interview asked how many families had their own pastureland, but because pastures are public property in Kyrgyzstan, the statistics reveal how many households used pasture for grazing and paid pasture ticket costs to pasture committees. Only Ak-Tyuz complains of a lack of pasture since it has no access to it. When it comes to pastures, Duisheevsky has the best adaptive capacity value. This is because its residents have access to relatively large grazing grounds.

### 4.2. Sensitivity

All four village districts have roughly comparable sensitivity levels, with Ak-Tyuz having the greatest overall sensitivity value at 0.32, Duisheevsky at 0.26, and Kok-Oirok and Chon-Kemin at 0.25 (Table 6). Despite the fact that health sensitivity scores are quite low, the nearest health clinics for herders are rather far away. There are designated first-aid stations in the communities, but dealing with more serious circumstances is challenging. Almost all homes reported problems with clean drinking water and access to private toilets. While there are few disputes over accessible water, and available water for cattle presents little concern, the sanitary condition shows to be a source of potential risk. Several studies in South-East Asia point to the relevance of personal cleanliness and sanitation in

combating disease transmission and outbreaks in rural regions [14,27]. Overall, agricultural and food security sensitivity is low since the Chui area has a good environment and market system for cultivating agricultural commodities. Over the last few decades, the number of cattle in Kyrgyzstan has continuously increased, while reports of excessive pasture degradation and overgrazing have been published [29]. However, the expanding number largely relates to wealthy houses and farms that expand their animal herds, whereas regular households cannot afford it. As a result, the aggregate statistics are extremely low. Human–wildlife conflict sensitivity is extremely low, particularly in the case of snow leopards. There have been no recent incidents of conflict with snow leopards, and the two cases cited by former herders during the field research occurred between 15 and 20 years ago. The scenario is slightly different with wolves, where typical livestock losses in the event of a human–wildlife conflict varied from 5 to 15 units. In addition to human–wildlife interactions, herders noted weather circumstances that caused them to lose livestock units. Natural resource sensitivity is particularly high since all four village districts rely substantially on natural resources. Agriculture is their primary source of income, with little to no diversification, as discussed in the preceding sub-section on adaptive capability. Livelihoods rely on farming and animal husbandry, making herders vulnerable to climate-change-related natural resource difficulties [14], especially given the existing overuse of pastures [29]. Because of a subsidized rate regime for power, the sensitivity to energy is quite low. Animal dung is commonly used to provide energy for cooking, which might have detrimental health consequences owing to carbon emissions [27].

### 4.3. Exposure

Climate change and disaster vulnerability is shown to be quite minimal in all four village areas. Droughts and snowstorms are the only natural calamities reported in the past five years. However, the rising temperature trends described previously in this research suggest that if the climate changes, exposure sensitivity may increase. During their research in India, Venus et al. saw similar results [27].

### 4.4. Human–Wildlife Conflict

Among one of the most protected species in Kyrgyzstan is the snow leopard (*panthera uncia*). To enable the survival of this species, any form of human–wildlife conflict needs to be eliminated. Based on the desktop review, the main threats for snow leopard survival in Kyrgyzstan identified are as follows:

- Decrease in habitat due to livestock grazing, which often leads to land degradation;
- Decrease in prey species due to uncontrolled hunting and livestock grazing;
- Climate change affecting habitat area and location;
- Touristic, mining, and other human-induced activities;
- Lack of capacities to monitor and observe wildlife.

There have been no known cases of snow leopard killings in recent decades; however, there have been a few cases of livestock being killed by snow leopards on a national scale. This information could be confirmed during the field visits and LVI assessment in 2022 as described in the sensitivity section. While the overall amount of human–wildlife conflicts are very low, including no recent ones with snow leopards in the pilot region, there were a few incidents regarding wolves [17,30].

## 5. Conclusions and Recommendations

This assessment analyzed the livelihood vulnerability of herders in the pilot region of Chui Oblast in Kyrgyzstan. Sensitivity, adaptability, and exposure were the main categories of analysis, where the dependency on natural resources showed the highest vulnerability in all four pilot village districts. Climate change and disaster vulnerability was relatively low, with extreme weather events such as droughts and snowstorms being the only climate-related disasters mentioned. With rising temperatures and more extreme changes, this delicate system, which is so dependent on the availability of natural resources, it is likely

that the exposure sensitivity will increase. The required actions to secure pastoralists' livelihoods and ensure sustainable land use at the same time are manifold and long. If the available resources are managed and monitored responsibly, the exposure sensitivity might not increase dramatically over the course of rising temperatures. However, the situation is fragile, and a diversification of income sources may contribute to decreased headers vulnerability towards climate change in Kyrgyzstan. Kyrgyz NGO CAMP Alatoo, widely working in area sustainable natural resource management and biodiversity conservation, has the scientific-based argumentation provided by the LVI study to stimulate urgent action towards environmental issues, such as the degradation of resources and projected climate change impact on the household's vulnerabilities. This will be communicated in CAMP Alatoo's interactions with state institutions, such as the Pasture Department of the Ministry of Agriculture, Forest Service and Biodiversity department and the Ministry of Ecology, Natural Resources and Technical Supervision.

In general, we received good standard LVI and IPCC framework results, with both confirming low sensitivity and exposure factors. This might be highly encouraging if the adaptive capacity is very strong, but it also demonstrates that present livelihood vulnerability indices are very delicate and unstable. The herders' heavy reliance on natural resources, tired status, lack of livelihood diversification choices, and projected climate change impact could all have a negative influence on them. In Kyrgyzstan, hundreds of livestock and dozens of wild animals died as a result of late frosts in pastures in 2022. Based on CAMP Alatoo's experience in sustainable natural resource management [29,31] and biodiversity conservation for improving livelihoods since 2004, as well as the survey results, the following management recommendations are proposed:

1.  Development of state compensation procedures for losses caused by natural disasters. There are no programs or systems in place to help herders cope with the dangers posed by weather extremes and related disasters. This, together with other adaptive capacity considerations, makes livelihoods extremely fragile. Cattle may perish not only as a result of climate disasters, but also as a result of a lack of feed, blocked highways, and other factors triggered by these calamities. In June 2022, 20 argali and 2300 heads of livestock died in northern Kyrgyzstan due to severe snowfall that limited access to grass;

2.  Insurance programs/mechanisms for livestock losses caused by snow leopard attacks are being developed. Extensive efforts are being made to conserve and expand snow leopard populations. Gradually, the programs' actions are yielding positive outcomes. Local villages began reporting occurrences of snow leopard attacks in recent years, which had not previously occurred in Kyrgyzstan. To avoid potential HWCs, mitigating risks should be in place, which may include not only snow leopards but also bears, wolves, and other species;

3.  Stakeholder capacity enhancement in wildlife monitoring, biodiversity research, and pasture management. Natural resource degradation makes ecosystems and biodiversity more susceptible, particularly in the context of climate change [29,31]. However, there is an institutional gap in the preparation of skilled specialists in natural resource management and biodiversity protection, including scientific observation and study, at the national level. As a result, the SPNAs' staff has limited capacity and capability for effective and sustainable management;

4.  Protection work is inadequately implemented due to the poor capacity of protected areas' personnel, as well as a lack of understanding among the local population about the need for biodiversity conservation and natural resource restoration. Raising local people's understanding of sustainable natural resource usage in the context of climate change and rising human effect on the environment will also be critical [29].

5.  Improve collaboration and communication among all relevant stakeholders and officials for improved livelihoods and biodiversity conservation. Create activities and collaboration mechanisms that improve access to information from local authorities, park management, the hydrometeorological service of the Kyrgyz Republic's Ministry

of Emergency Situations, and so on. This should allow stakeholders to improve LVI computation and prognostic research and scenario design.

**Author Contributions:** Conceptualization, S.W.W.; methodology, S.W.W.; software, S.W.W. validation, S.W.W. and A.I.; formal analysis, A.I., M.Z. and S.W.W.; investigation, A.I. and S.W.W.; resources, S.W.W. and W.-K.L.; data curation, A.I. and S.W.W.; writing—A.I. and S.W.W.; writing—review and editing, S.W.W. and A.I; visualization, A.I. and S.W.W.; supervision, S.W.W. and W.-K.L.; project administration, S.W.W.; funding acquisition, S.W.W. and W.-K.L. All authors have read and agreed to the published version of the manuscript.

**Funding:** This research was supported by the Core Research Institute Basic Science Research Program through the National Research Foundation of Korea (NRF) funded by the Ministry of Education (NRF-2021R1A6A1A10045235).

**Data Availability Statement:** Data is available from the corresponding author. Email: wangsonam@korea.ac.kr.

**Acknowledgments:** To the research team of CAMP Alatoo–Ruslan Ismailov, Barchynbek Meimanbekov and Kanat Zhuzubaliev; data analysis team-interns at CAMP Alatoo: Jan Austen and Johanna Antretter; editing by Helen Koch. The authors would like to thank the National Research Foundation of Korea and the OJEong Resilience Institute at Korea University for supporting this study. In addition, we would also like to thank the nomads for voluntarily participating in this study.

**Conflicts of Interest:** The authors declare no conflict of interest.

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
