# Peer review of "Assessing the Livelihood Vulnerability of Herders to Changing Climate in Chui Oblast, Kyrgyz Republic"

_land, doi:10.3390/land12081520_

Round 1
Reviewer 1 Report
The manuscript is clear, relevant to the field, and presented in a well-structured format. The manuscript is scientifically sound and is the experimental design appropriate to the hypothesis.
Introduction
The introduction is very well-articulated but lacks citations from the references used. The citations needed are specified in the attached manuscript. This review is a requirement for publication.
Methodology
The methodology was very well written. The experimental design was appropriate to test the hypothesis. The results are reproducible based on the details given in the methods.
Results
Results are well presented, quite rich, and fulfilled the article’s objectives. The separation and 3.1 Overview l and 3.2 Subcomponents made the results much clear. Tables and graphs are well designed, although tables are a bit too long and badly designed.
Discussion
The discussion is well structured; however, a few references need to be updated.
Conclusion
As marked in the attached file, most of your conclusion is part of the discussion, especially those paragraphs with citations. Some of it could be even a “Further Recommendation” session, but not a conclusion
Your conclusion needs to be rewritten. In a good conclusion, you should only restate the thesis and show how it has been developed through the body of the paper. Briefly summarize the key arguments made in the body, showing how each of them contributes to proving your thesis
“The Conclusion is an important part of your paper where you distill your study and give the paper a sense of finality. A good Conclusion section encourages a reader to appreciate your work in light of the bigger picture. It should be brief and concise”
References
A few need to be updated.

Reviewer 2 Report
Global warming has gradually become the human consensus. Climate change will have an important impact on the livelihoods of herders. Based on first-hand survey data, the authors constructed an index system to measure herders' livelihood vulnerability to climate change. Overall, the topic selection makes some sense, but the study design is too crude to be accepted in the current version. Specific suggestions are as follows:
(1) The introduction needs a modest rewrite. One is that it is not clear what the key scientific problem the eye is solving. Second, compared with the existing research, the marginal contribution of this research is not clear. In fact, as far as I know, there are a large number of studies on measuring livelihood vulnerability to climate change. Compared with these studies, it is not clear where the marginal contribution of this study is. In addition, the reason for the focus on the pastoral household and the uniqueness of the pastoral household compared to other small farmers need to be further explained.
(2) The lack of rigorous theoretical analysis framework makes the selection of research indicators lack of theoretical basis, and it is difficult to be persuasive. At the same time, the author constructs a set of index system to measure herders' livelihood vulnerability to climate change. The selection of indicators is directly related to the subsequent results, and the results of indicator changes will also change. Because of the lack of a frame of reference, the results are of little practical significance.
(3) At present, only 18 references are cited in this paper, which is not desirable for the present situation where there are already a large number of references.
Minor editing of English language required.
Round 2
Reviewer 2 Report
I have no other comments, thank you.
Author Response
Thank you for your updated review, well received.